# Family Socioeconomic Status and Adolescent Mental Health Problems during the COVID-19 Pandemic: The Mediating Roles of Trait Mindfulness and Perceived Stress

**DOI:** 10.3390/ijerph20021625

**Published:** 2023-01-16

**Authors:** Yue Yuan, Aibao Zhou, Manying Kang

**Affiliations:** 1Department of Psychology, Northwest Normal University, Lanzhou 730000, China; 2Department of Social Work, Hong Kong Baptist University, Kowloon Tong, Kowloon 999077, Hong Kong

**Keywords:** family socioeconomic status, mental health, trait mindfulness, perceived stress, adolescence

## Abstract

The present study was conducted twice over one year during the COVID-19 pandemic with Chinese adolescents (*n* = 1582) to investigate the relationships among family socioeconomic status (SES), adolescent mental health problems, trait mindfulness, and perceived stress using self-reported measures. We administered the Mindful Attention Awareness Scale (MAAS), the Perceived Stress Scale (PPS), the Self-rating Anxiety Scale (SAS), the Epidemiologic Studies Depression Scale (CES-D), and Conduct Problem Tendency Inventory (CPTDI) to a sample of Chinese adolescents. The results prove that (1) there were significant correlations among socioeconomic status, trait mindfulness, perceived stress, and adolescent mental health problems, and the (2) serial mediation analysis indicated that trait mindfulness and perceived stress performed as mediators on the path from SES to anxiety, depression, and externalizing problems. Our findings provide a contribution by showing the connection between socioeconomic position and adolescent mental health problems and by offering a reference for the treatment of psychological issues affecting adolescents.

## 1. Introduction

The 2019 coronavirus disease (COVID-19) is a worldwide pandemic. The Chinese government took decisive action to seal off the city in order to prevent the spread of the contagious virus to other areas. This included school closures, work suspensions, production halts, and community containment [1], which had a negative impact on most families’ income. During the COVID-19 outbreak, it is estimated that 9.0% to 53.5% of students globally experienced psychological distress symptoms [2,3]. However, little is known about the relationship between family socioeconomic status (SES) and mental health problems among Chinese adolescents [3,4]. Thus, the first contribution of our research was to investigate the impact of SES on student mental health problems during the COVID-19 pandemic in a longitudinal perspective.

The COVID-19 pandemic’s widespread influence on depression was found to be 26.9%, on anxiety was found to be 21.8%, and on stress-related symptoms was found to be 48.1%, which has a negative influence on an individual’s mental health [4]. Adolescents with anxiety and depression are less motivated to study, have difficulty concentrating in class, and have bad academic performance [5]. In addition, adolescents frequently engage in maladaptive behaviors that externalize difficulties, such as aggressive conduct [6]. However, to our knowledge, few studies have simultaneously investigated the impact of family socioeconomic status on adolescent anxiety, depression, and externalizing problems during pandemic. The second contribution of our research is that by investigating the psychological factors that influence adolescent mental health, we can gain a more nuanced understanding of adolescent mental health and well-being and potentially gain insight into interventions that promote more resilient trajectories for the benefit of adolescents and the broader community during the COVID-19 pandemic.

### 1.1. Socioeconomic Status and Adolescent Mental Health Problems

Family socioeconomic status (SES) is a ranking based on owned and controlled the combination of economic capital, social relationship capital, political capital, and cultural capital by the family [7]. Numerous studies have demonstrated a strong link between an individual’s initial SES and childhood experiences as well as future development [8]. Adolescents from different family monetary backgrounds have different perceptions and attributions of unfair treatment. A high SES provides adolescents with adequate social support and subsequently can mitigate or prevent adolescents from developing internalizing and externalizing problems [5]. In contrast, adolescent with low SES may experience physical and psychological destruction due to a lack of parental nurturing and patient guidance [5].

The COVID-19 outbreak brought a negative impact on families’ income because some parents lost their jobs or were suspended from work. There are fewer studies on whether the impact of the epidemic on family economic income has an impact on adolescent mental health. In view of this, our research tried to investigate the association between SES and adolescent mental health problems. In addition, we hypothesize that SES showed a negative impact on adolescent mental health problems during the COVID-19 pandemic (H1).

### 1.2. Socioeconomic Status and Adolescent Mental Health Problems: Role of Trait Mindfulness

Mindfulness training intervention has a positive influence on resilience, emotional regulation, and psychological capital during the COVID-19 pandemic [9,10], because mindfulness allows individuals to pay thoughtful attention to and be aware of unforeseen events as they occur [11]. Mindfulness can be increased through training, and it also can be thought of as a psychological trait that relates to the inclination to be mindful in everyday life [9].

Parenting style influences an individual’s formation of trait mindfulness [12,13]. For instance, Ryan et al. (2007) reported different attachment types and experiences of infants as one of the reasons that result in different trait mindfulness between individuals [14]. Conversely, researchers traced individuals with low-level trait mindfulness and his/her childhood experiences and have discovered they frequently experienced insecure parent-child attachments, such as emotional neglect or abuse [15]. Research on adolescents found that a warm attachment was positively associated with trait mindfulness, while an insecure attachment was negatively associated with low adolescent trait mindfulness [13]. Empirical studies on adolescents showed that parent trait mindfulness affects adolescent mental health [12,16].

SES has an impact on parenting style [15,17]. Individuals with lower SES experience a disproportionate distribution of social stress. Compared to their wealthier peers, poor adults experience more stressful lives, including, but not limited to, poorer housing quality, greater congestion and noise levels, and more day-to-day commitments [17]. In addition, parents with low SES typically experience more financial stress, family conflict, domestic violence, and discrimination [18]. These negative events are more likely to lead to negative parenting and predispose them to form insecure attachments [15], leading to low trait mindfulness.

However, to our knowledge, there has been little direct research to prove that trait mindfulness is related to SES yet. Based on those statements, our second hypothesis is that trait mindfulness mediates the relationship between SES and adolescent mental health problems (H2). Therefore, the third contribution of our research was to investigate the role of trait mindfulness via SES to adolescent mental health problems to provide a theoretical basis in the intervention of mental health problems in families with low SES adolescents caused by the COVID-19 pandemic.

### 1.3. Socioeconomic Status and Adolescent Mental Health Problems: Role of Mindfulness and Perceived Stress

Perceived stress is a response when an individual encounters challenging context requirements [11]. Trait mindfulness has positive factitive effects on stress management. The reperception model of mindfulness suggests that mindfulness can help individuals to reperceive moment-to-moment experiences more objectively, remove unconscious behavioral and emotional patterns, and promote adaptive responses to negative stimuli [19]. Thus, high levels of mindfulness may have a protective effect against stress [11,20].

Several studies have shown that perceived stress has a positive predictive effect on mental health problems, such as hostility [21,22] and depression [23,24,25]. Pronounced stress reactions are characteristic of individuals who score highly on measures of trait hostility [21]. However, Nykliček et al. (2013) stated mindfulness training reduced blood pressure responses in acute stressful situations [26]. Awareness of the present moment in mindfulness helps individuals cope with stress in their daily lives [26]. Generally, the impact of a particular stress varies across individuals. As a result, the likelihood of a depressive episode is hypothesized to increase when individuals perceive stress as uncontrollable, unpredictable, and severe and deem coping resources as insufficient [23].

These works have revealed that perceived stress and mindfulness are responsible for mental health problems. Hence, our third hypothesis is that trait mindfulness and perceived stress play a chain-mediating role in the relationship between SES and mental health problems (H3). Thus, the fourth contribution of our research hopes to uncover the relationship among SES, perceived stress, trait mindfulness, and mental health problems during the COVID-19 pandemic and to provide an empirical reference for the intervention of mental health problems caused by the COVID-19 pandemic.

## 2. Materials and Methods

### 2.1. Participants

We adopted a whole-group sampling method; students from a junior high school were selected as participants and administered as a group in a class. Our research was approved by the Research Ethics Committee at our university and the leaders of the junior high school. In September 2021, we distributed 2144 questionnaires to explore adolescent family socioeconomic status, trait mindfulness, and perceived stress. After deleting consistent response questionnaires, 2086 valid questionnaires, with a valid return rate of 97.29%, were obtained. In November 2022, we issued 1604 questionnaires to participants to assess their anxiety, depression, and externalizing issues. Every questionnaire was made available on https://www.wjx.cn/ (accessed on 12 January 2023). The number of valid surveys was 1582, with an efficiency rate of 98.63%, after eliminating invalid questionnaires with consistent answers. The variables of interest were collected from the same participants at two different times. All adolescents and their parents completed the informed consent form.

In order to measure power, we used maximum likelihood estimation in Mplus 8.0 [27]. Our sample size was adequate to identify modest effects in RI-CLPMs and regular CLPMs with power >0.80, according to an analysis of Monte Carlo power [28]. After that, we used SPSS 26.0 for statistical analysis. Pearson’s correlation analysis was conducted to test the association among all variables. Gender and age were the control variables in the analysis. The participants’ demographic information listed in Table 1.

### 2.2. Measurement

#### 2.2.1. Family Socioeconomic Status (SES)

Internationally, occupational prestige is one of the indicators of social capital, but social capital also exists in schools and communities [29]. The classification of occupation in China has been controversial, and individuals with the same occupation have a large gap in income and education level [30]. In addition, Bornstein et al. (2003) pointed out that using occupation as an indicator of socioeconomic status of a low-income population is not reliable enough; thus, occupation was not included as an indicator of SES in this research [31]. In summary, SES in this study was measured using Liang, Wang, and Yu (2021), synthesizing from the parents’ educational attainment and average monthly household income [32]. SES in our research including parents’ education: (1 = junior high school and below, 2 = high school, 3 = college/bachelor’s degree, and 4 = master’s degree and above) and average annual household income: (1 = CNY 15,000 or less, 2 = CNY 15,000 to CNY 30,000, 3 = CNY 30,000 to CNY 60,000, 4 = CNY 60,000 to CNY 100,000, 5 = CNY 100,000 to CNY 200,000, and 6= CNY 200,000 or more). The calculation formula is: Z_SES_ = Z_father’s education_ + Z_mother’s education_ + Z_monthly household income_.

#### 2.2.2. Mindful Attention Awareness Scale (MAAS)

Trait mindfulness was assessed with the Chinese version of the Mindful Attention Awareness Scale (MAAS), which is composed of 15 items [33]. Participants indicate how often statements apply on a 6-point Likert scale (1 = applies to me very much or most of the time, 6 = does not apply to me at all), which was adopted for the purpose of the present study [33]. Higher scores indicate higher levels of mindfulness and the more present-attention and awareness traits. The MAAS has consistently shown excellent psychometric properties with high levels of construct validity and reliability [33,34]. The reliability is 0.93 in this study.

#### 2.2.3. Perceived Stress Scale (PSS)

Chinese Version of the Perceived Stress Scale (PSS), revised by Yang and Huang (2003), was used to measure stress [35]. The scale consists of 14 items, including 2 dimensions of tension (7 items) and loss of control (7 items). The scale is scored on a 5-point Likert scale (1–5), with 1 = “never” and 5 = “a lot”. The scale is used to assess the level of stress an individual has experienced in the past month, with higher scores indicating greater perceived stress. In this study, the Cronbach’s alpha coefficient for this scale was 0.75.

#### 2.2.4. Mental Health Problems: Anxiety

We measured anxiety with the Self-rating Anxiety Scale (SAS) revised by Tao et al. (1994), which consists of 20 items [36]. Self-rating Anxiety Scale is scored on a 4-point Likert scale, with 1 = “does not apply to me at all” and 5 = “applies to me very much or most of the time”. Overall, the higher the score, the more pronounced anxiety. Cronbach’s alpha of the scale is 0.77 in our study.

#### 2.2.5. Mental Health Problems: Depression

Depression was measured with 10-item Center for Epidemiologic Studies Depression Scale (CES-D; Chinese form: Zhang and Li, 2011). Responses are given on a 4-point Likert Scale, with 1 = “I do not agree at all”, 4 = “fully agree”. Higher scores indicate higher levels of depression [37]. In our sample, the Cronbach alpha is 0.83.

#### 2.2.6. Mental Health Problems: Externalizing Problems

Adolescents externalizing problems was evaluated via Conduct Problem Tendency Inventory (CPTDI) organized by Zhang et al. (2009). Conduct Problem Tendency Inventory includes a 14-item list and comprises three dimensions of violation tendency, addiction tendency, and aggression tendency [38]. Participants were instructed to select how often they experienced each effect in general via a five-point Likert sale from 1 (does not apply to me at all) to 5 (applies to me very much or most of the time). The reliability was 0.84 in the present study.

### 2.3. Data Analysis

Descriptive statistics and preliminary analysis were performed using SPSS 26.0 software (IBM, Armonk, NY, USA). The PROCESS macro for SPSS version 3.4 developed by Hayes (2018) was chosen for the serial mediation analysis in our current research [39]. Particularly, bias-corrected confidence interval (95%) based on 5000 bootstrap replications was used to determine the significance of the indirect effect.

### 2.4. Common Method Biases Test

Because our data are based on self-reporting, there might be common method biases. We performed Harman one-way method to test common method biases test for interindividual variables. Exploratory factor analysis was conducted on all items (a total of 76), and 16 factors with characteristic roots greater than 1 were obtained, among which the largest factor only explained 21.00% of the total variation. It indicates that there is no serious common method bias in the data of this research.

## 3. Results

### 3.1. Descriptive Statistics and Correlations

The correlations, means, standard deviations, and alpha coefficients for socioeconomic status (SES), trait mindfulness, perceived stress, anxiety, depression, and externalizing problems are demonstrated in Table 2, which indicates high internal consistency of measures and relationships in the expected direction. We used Pearson’s correlation analysis. The correlation between SES and trait mindfulness is positively significant; the correlation between SES and perceived stress is negatively significant; the correlation between SES and anxiety is negative significant; the correlation between SES and depression negatively significant; the correlation between SES and externalizing problems is negative significant. The correlation between trait mindfulness and perceived stress is negatively significant; the correlation between trait mindfulness and anxiety is negatively significant; the correlation between trait mindfulness and depression is negatively significant; the correlation between trait mindfulness and externalizing problems is negatively significant. The correlation between perceived stress and anxiety is positively significant; the correlation between perceived stress and depression is positively significant; the correlation between perceived stress and externalizing problems is positively significant. The correlation between anxiety and depression is positively significant; the correlation between anxiety and externalizing problems is positively significant. The correlation between depression and externalizing problems is positively significant.

### 3.2. Path Analysis

The results of the regression analysis indicate (Table 3) that SES positively predicted trait mindfulness, with (β = 0.07, SE = 0.01, t = 4.94, *p* < 0.01, 95% CI [0.04, 0.10]). Trait mindfulness negatively predicted perceived stress, with (β = −0.28, SE = 0.01, t = −21.58, *p* < 0.01, 95% CI [−0.31, −0.26]); SES negatively predicted perceived stress, with (β = −0.02, SE = 0.01, t = −1.89, *p* = 0.058, 95% CI [−0.03, −0.01]). After including trait mindfulness and perceived stress in the regression equation, SES did not predict anxiety, with (β = −0.01, SE = 0.005, t = −0.46, *p* > 0.05, 95% CI [−0.01, 0.01]). Perceived stress positively predicted anxiety, with (β = 0.30, SE = 0.02, t = 17.61, *p* < 0.001, 95% CI [0.27, 0.34]). Trait mindfulness negatively predicted anxiety, with (β = −0.08, SE = 0.01, t = −8.03, *p* < 0.001, 95% CI [−0.10, −0.06]). Trait mindfulness negatively predicted depression, with (β = −0.15, SE = 0.01, t = −11.48, *p* < 0.001, 95% CI [−0.17, −0.12]). In addition, perceived stress positively predicted depression, with (β = 0.51, SE = 0.02, t = 23.59, *p* < 0.001, 95% CI [0.47, 0.56]). However, the effect of SES on depression was not significant, with (β = −0.004, SE = 0.007, t = −0.59, *p* > 0.05, 95% CI [−0.02, 0.01]). Trait mindfulness negatively predicted externalizing problems, with (β = −0.40, SE = 0.02, t = −24.86, *p* < 0.001, 95% CI [−0.43, −0.37]). Perceived stress positively predicted externalizing problems, with (β = 0.31, SE = 0.03, t = 11.55, *p* < 0.001, 95% CI [0.25, 0.36]). Lastly, SES negatively predicted externalizing problems, with (β = −0.03, SE = 0.01, t = −3.35, *p* < 0.01, 95% CI [−0.04, −0.02]).

Our result indicate that trait mindfulness and perceived stress play a chain-mediating role on the path from SES to mental health problems. Trait mindfulness and perceived stress play serial mediators on the path between SES and anxiety and depression. Notably, the direct effect of SES on anxiety and depression was insignificant, while the indirect effect was significant after adding the mediators of trait mindfulness and perceived stress. In addition, the direct effect of SES on externalizing problems was significant. The indirect effect of SES on externalizing problems via trait mindfulness and perceived stress was also significant.

### 3.3. Serial Multiple Mediation Model

In order to examine the serial multiple mediation model of SES to mental health via trait mindfulness and perceived stress, we tested three possible mediating pathways linking SES to anxiety, depression, and externalizing problems, which led to three models. The first model path is SES to anxiety via trait mindfulness to perceived stress. The second model path is SES to depression via trait mindfulness to perceived stress. The third model path is SES to depression via trait mindfulness to externalizing problems.

#### 3.3.1. Serial Multiple Mediation Model: Anxiety

In the first model (Figure 1), the total model effect was significant (R = 0.54, R^2^ = 0.30, F (3, 1579) = 223.84). The direct effect of SES on anxiety was insignificant (β = −0.01, SE = 0.005, t = −0.46, 95% CI [−0.013, −0.008]). The indirect effect of SES on anxiety was significant (β = −0.003, SE = 0.05, 95% CI [−0.13, 0.01]), including the path from SES via trait mindfulness to anxiety (β = −0.006, SE = 0.002, 95% CI [−0.010, −0.003]); the serial path from SES via trait mindfulness to perceived stress to anxiety (β = −0.006, SE = 0.001, 95% CI [−0.01, −0.004]); and the path from SES via perceived stress to anxiety (β = −0.05, SE = 0.003, 95% CI [−0.01, 0.004]), as can be seen in Table 4.

#### 3.3.2. Serial Multiple Mediation Model: Depression

In the second model (Figure 2), the total model effect was significant (R = 0.66, R^2^ = 0.44, F (3, 1579) = 416.82). The direct effect of SES on depression was insignificant (β = −0.04, SE = 0.007, 95% CI [−0.02, 0.01]). The indirect effect of SES on depression was significant (β = −0.03, SE = 0.01, 95% CI [−0.04, −0.02]), including the path from SES via trait mindfulness to depression (β = −0.011, SE = 0.02, 95% CI [−0.02, −0.01]); the serial path from SES via trait mindfulness to perceived stress to depression (β = −0.011, SE = 0.002, 95% CI [−0.02, −0.01]); and the path from SES via perceived stress to depression (β = −0.008, SE = 0.01, 95% CI [−0.02, −0.01]), as can be seen in Table 4.

#### 3.3.3. Serial Multiple Mediation Model: Externalizing Problems

The serial multiple mediation model of SES to externalizing problems via trait mindfulness and perceived stress is shown in Figure 3. The total model effect was significant, (R = 0.68, R^2^ = 0.47, F (3, 1579) = 465.06). The direct effect of SES on externalizing problems was insignificant (β = −0.03, SE = 0.01, t = −3.35, 95% CI [−0.04, −0.01]). The indirect effect of SES on externalizing problems was significant (β = −0.04, SE = 0.008, 95% CI [−0.06, −0.03]), including the path from SES via trait mindfulness to externalizing problems (β = −0.03, SE = 0.006, 95% CI [−0.04, −0.02]); the serial path from SES via trait mindfulness to perceived stress to externalizing problems (β = −0.01, SE = 0.002, 95% CI [−0.001, −0.004]); and the path from SES via perceived stress to externalizing problems (β = −0.005, SE = 0.003, 95% CI [−0.01, −0.003]), as can be seen in Table 4.

## 4. Discussion

The new contribution of our research is the discovery of the relationship among adolescent SES, trait mindfulness, perceived stress, and mental health problems during the COVID-19 pandemic. The results demonstrate that there was a significant relationship among SES, trait mindfulness, perceived stress, and mental health problems, including anxiety and depression and externalizing problems.

Another contribution of our study is that we uncovered that trait mindfulness and perceived stress play a chain-mediating role on the path from SES to mental health problems. For internalizing problems, trait mindfulness and perceived stress play serial mediators on the path between SES and anxiety and depression. Notably, the direct effect of SES on anxiety and depression was insignificant, while the indirect effect was significant after adding the mediators of trait mindfulness and perceived stress. Secondly, the direct effect of SES on externalizing problems was significant. The indirect effect of SES on externalizing problems via trait mindfulness and perceived stress was also significant.

### 4.1. The Relationship between SES and Mental Health Problems

As our hypothesis 1 expected, there was a significant association between SES and adolescent mental health problems. The family stress model provides a rationalization for this result; family economic difficulties and the resulting family pressure will lead to higher psychological pressure of the parents, which in turn leads to negative emotions in adolescents [40]. For example, a longitudinal study found that economic stress in an individual’s early life predicted depressive symptoms [41]. SES has been shown to have a significant influence on adolescent physical, mental, and social cognition beginning before birth and extending into adulthood [42]. Family economic difficulties can enhance the risk of depression, anxiety, and aggression, which can gravely hinder an individuals’ socioemotional development and negatively affect their mental health [8]. Due to the irregular gap between the affluent and the poor in society, SES has become an important social influence that is significantly associated with adolescent mental health problems.

In addition, both social class cognitive theory and empirical studies indicate that family financial difficulties are associated with problem behaviors, such as depression, anxiety, and aggression. In addition, family economic difficulties are factors that hinder adolescent social–emotional development, thereby increasing an individuals’ risk of developing problem behaviors, thus negatively impacting adolescent mental health [43,44]. Family investment theory indicates that adolescents with high family SES have more social resources than adolescents with a low family SES, and, accordingly, adolescents with low family SES encounter more stress and hardship growing up, leading to the experience of more negative emotions and behavioral problems [40].

### 4.2. Independent Mediating Effect of Trait Mindfulness

Our results show that the indirect effect of SES on anxiety, depression, and externalizing problems via the mediation of trait mindfulness were significant and indicate that trait mindfulness mediates the relationship between SES and adolescent mental health problems. Hence, hypothesis 2 was verified. Specifically, SES positively predicted trait mindfulness in the first path, while trait mindfulness negatively predicted anxiety, depression, and externalizing problems. Trait mindfulness is a beneficial factor in decreasing mental health problems.

Adolescents with an elevated SES correlate with high trait mindfulness that may be due to parents with an elevated SES and experience a less disproportionate distribution of social stress. Compared to wealthier parents, poor parents experience more stressful lives, including, but not limited to, poorer housing quality, greater congestion and noise levels, and more day-to-day commitments [17]. These surroundings lead to different parenting styles. Nonetheless, the parenting style influences an individual’s formation of trait mindfulness [12,14]. For instance, children with a secure attachment type always accompanied by a tight relationship with parents exhibit high trait mindfulness levels [45]. At the same time, theoretical studies of the parental acceptance–rejection theory [46] point out that an individual’s development is deeply influenced by family. Children who often experience acceptance during their early interactions with their parents or caregivers are more likely to develop an accepting attitude toward the world. An accepting attitude is part of trait mindfulness [11], which leads to higher levels of mindfulness and vice versa. Therefore, SES negatively predicts adolescent mental health problems via the mediation of trait mindfulness.

### 4.3. Serial Mediating Effect of Trait Mindfulness and Perceived Stress

The results of this research show that the chain-mediating effect of trait mindfulness and perceived stress on the influence of SES on adolescent mental health problems was significant. Thus, research hypothesis 3 is valid. SES positively predicted trait mindfulness and negatively predicted perceived stress, which means adolescents with high SES were accordingly associated with high trait mindfulness and low perceived stress and vice versa. Adolescents with low SES accompanied with low trait mindfulness, perceived more stress, and ultimately developed mental health problems. On the contrary, adolescents with high SES accompanied with high trait mindfulness, perceived less stress, and eventually had fewer chances to engage in internalizing or externalizing problems.

This might because SES has an impact on parenting style [17], which influences the formation of adolescent trait mindfulness [12,14]. Research on adolescents found that a warm attachment was positively associated with high levels of mindfulness, while an insecure attachment was negatively associated with adolescent mindfulness [13]. Parents with low SES typically experience more financial stress, family conflict, domestic violence, and discrimination [18]. These negative events are more likely to lead to negative parenting; thus, parenting with negative emotions can predispose them to form insecure attachments, leading to low mindfulness [15].

Trait mindfulness negatively predicted perceived stress, and this was consistent with previous studies. A study on law enforcement officers demonstrated that individuals higher in mindfulness reported lower perceived stress [47]. Individuals with lower SES experienced a disproportionate distribution of social stress. Compared to their wealthier peers, poor youth and adults experience more stressful lives, including, but not limited to, poorer housing quality, greater congestion and noise levels, and more day-to-day commitments [7].

The stress-buffering theory proposes that mindfulness can act as a buffer against stress and reduce the adverse consequences of stress for individuals [48]. Some research utilizes mindfulness-based stress reduction intervention to decrease a participant’s perceived stress [49,50]. That is also supported by empirical research. For example, a study by Brown et al. (2012) discovered that individuals with high levels of trait mindfulness had lower levels of stress-induced cortisol activity in extreme stress situations [51]. In another instance, Creswell et al. (2014) illustrated a similar buffering effect, where mindfulness training was efficient in alleviating individuals’ negative evaluations of stress [52].

## 5. Conclusions

The findings of this study provide new perspectives on the prevention and intervention of mental health problems in adolescents during the COVID-19 pandemic. It was found that adolescents with low SES are prone to have mental health problems. In addition, trait mindfulness and perceived stress play a chain-mediating role. This means that in the process of preventing and intervening in the mental health problems of low-SES adolescents, the following aspects need to be considered:
1.The whole society should pay active attention to the development of low-SES adolescents, such as by establishing a sound welfare protection system for low-SES children during the COVID-19 pandemic.2.The chain-mediating effect suggests that perceived stress may lead to more mental health problems in low-SES adolescents. Therefore, teachers should learn as much as possible about students’ family socioeconomic backgrounds and pay timely attention to low-SES students when there is a large gap between the rich and the poor in the group during the COVID-19 pandemic.3.For adolescents with higher levels of stress perception, using interventions, such as Mindfulness-Based Cognitive Therapy, to prevent adolescents from internalizing the stress from SES into emotional problems or externalizing it into behavioral problems.4.For mental health professionals, training adolescents in perceived stress relief skills might prove effective for supporting mental health problems in the COVID-19 pandemic. In addition, to maximize their effectiveness, intervention programs should be developmentally sensitive to adolescent SES and level of stress relief competencies. Moreover, as trait mindfulness appears to influence mental health problems in adolescence, interventions should tackle this dimension first, which might have a snowball effect on the others.


### Limitations and Future Directions 

The first limitation of this pilot study is that our investigation of SES variables was based on objective adolescent reports. Future research could examine the impact of adolescent ratings of subjective SES on their mental health problems. The second limitation of this research is that all measures were derived from the self-reported scale, and future studies can combine various methods, such as interviews and other peoples’ evaluations to gain an in-depth understanding of the relationship among SES, trait mindfulness, perceived stress, and mental health problems. In the future, it may be more convincing to select a different sample from different cities if research resources and funds are sufficient.

## Figures and Tables

**Figure 1 ijerph-20-01625-f001:**
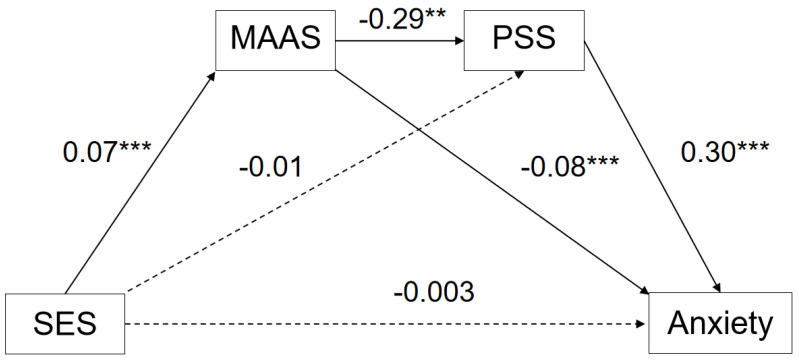
Serial Mediation Model of Socioeconomic to Internalizing Problems: Anxiety. ** *p* < 0.01. *** *p* < 0.001.

**Figure 2 ijerph-20-01625-f002:**
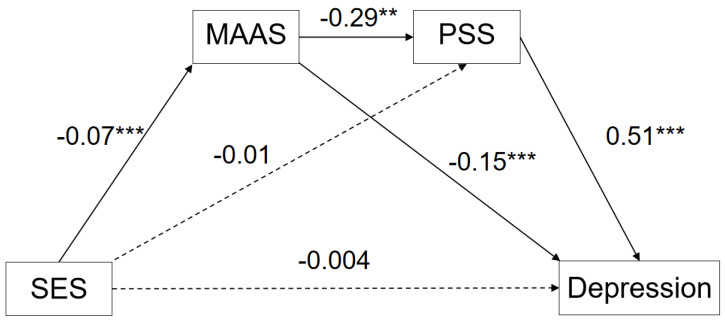
Serial Mediation Model of Socioeconomic to Internalizing Problems: Depression. ** *p* < 0.01. *** *p* < 0.001.

**Figure 3 ijerph-20-01625-f003:**
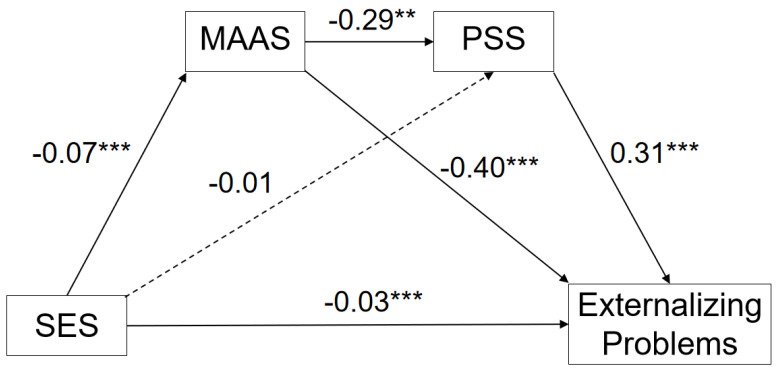
Serial Mediation Model of Socioeconomic to Externalizing Problems. ** *p* < 0.01. *** *p* < 0.001.

**Table 1 ijerph-20-01625-t001:** Demographic information of participants (*n* = 1582).

Variables		*n* (%)
Gender	Male	731 (46.18%)
	Female	852 (53.82%)
Grade	Tenth grade	570 (36.01%)
	Eleventh grade	522 (32.96%)
	Twelfth grade	491 (31.02%)
Age	14 years old	326 (20.59%)
	15 years old	490 (30.95%)
	16 years old	410 (25.90%)
	17 years old	357 (22.55%)
Father’s education	Elementary school and below	265 (16.75%)
	Junior high school	873 (55.18%)
	Senior/Vocational high school	390 (24.65%)
	Junior college	45 (2.84%)
	Bachelor’s degree	10 (0.63%)
Mother’s education	Elementary school and below	518 (32.74%)
	Junior high school	821 (51.90%)
	Senior/Vocational high school	229 (14.48%)
	Junior college	13 (0.82%)
	Bachelor’s degree	1 (0.06%)
	Master’s degree or above	1 (0.06%)
Annual household income	Less than CNY 15,000	348 (22.00%)
	CNY 15,000–30,000	608 (38.43%)
	CNY 30,000–60,000	319 (20.16%)
	CNY 60,000–100,000	199 (12.58%)
	CNY 100,000–200,000	87 (5.50%)
	CNY 200,000 above	22 (1.39%)

**Table 2 ijerph-20-01625-t002:** Correlations, means, and standard deviations of variables (*n* = 1583).

	1	2	3	4	5	6	*M*	*SD*
1. SES	1						−0.63	1.46
2. MAAS	0.12 **	1					3.12	0.86
3. PSS	−0.10 **	−0.48 **	1				1.99	0.52
4. Anxiety	−0.08 **	−0.40 **	0.52 **	1			1.90	0.37
5. Depression	−0.09 **	−0.49 **	0.63 **	0.68 **	1		1.98	0.52
6. EP	−0.15 **	−0.65 **	0.50 **	0.44 **	0.53 **	1	2.53	0.66

Note: ** *p* < 0.01; SES: family socioeconomic status; MAAS: trait mindfulness; PSS: perceived stress; EP: externalizing problems.

**Table 3 ijerph-20-01625-t003:** Analysis of the regression relationship among the variables.

Outcome	Predictor Variable	R	R^2^	F (df)	β	t
Trait Mindfulness		0.12	0.02	24.42 (1, 1581)		
	SES				0.07	4.94 ***
Perceived Stress		0.49	0.24	243.40 (2, 1580)		
	MAAS				−0.29	−21.58 ***
	SES				−0.01	−1.89
Anxiety		0.55	0.30	223.84 (3, 1579)		
	MAAS				−0.08	−8.00 ***
	PSS				0.30	17.61 ***
	SES				−0.002	−0.46
Depression		0.66	0.44	416.82 (3, 1579)		
	MAAS				−0.15	−11.48 ***
	PSS				0.51	23.59 ***
	SES				−0.004	−0.58
Externalizing Problems		0.68	0.47	465.06 (3, 1579)		
	MAAS				−0.40	−24.86 ***
	PSS				0.31	11.55 ***
	SES				−0.03	−3.35 ***

Note: SES: family socioeconomic status; MAAS: trait mindfulness; PSS: perceived stress; *** *p* < 0.001.

**Table 4 ijerph-20-01625-t004:** Analysis of mediation effect.

Dependent Variable	Path	*β*	*SE*	LLCI	ULCI
Anxiety	Direct Effect	−0.003	0.005	−0.01	0.01
	Total Indirect Effect	−0.02	0.002	−0.02	−0.01
	SES→MAAS→Anxiety	−0.01	0.002	−0.01	−0.003
	SES→MAAS→PSS→Anxiety	−0.01	0.001	−0.01	−0.004
	SES→PSS→Anxiety	−0.005	0.002	−0.01	−0.004
Depression	Direct Effect	−0.004	0.007	−0.02	0.01
	Total Indirect Effect	−0.03	0.006	−0.04	−0.02
	SES→MAAS→Depression	−0.01	0.002	−0.02	−0.01
	SES→MAAS→PSS→Depression	−0.01	0.002	−0.02	−0.01
	SES→MAAS→Depression	−0.01	0.01	−0.02	−0.001
Externalizing Problems	Direct Effect	−0.03	0.01	−0.04	−0.01
	Total Indirect Effect	−0.04	0.008	−0.06	−0.03
	SES→MAAS→EP	−0.03	0.006	−0.13	−0.08
	SES→MAAS→PSS→EP	−0.01	0.003	−0.01	0.003
	SES→MAAS→EP	−0.005	0.002	−0.01	−0.004

Note: SES: family socioeconomic status; MAAS: trait mindfulness; PSS: perceived stress; EP: externalizing problems.

## Data Availability

Data are available on request.

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
