# Peer review of "Family Socioeconomic Status and Adolescent Mental Health Problems during the COVID-19 Pandemic: The Mediating Roles of Trait Mindfulness and Perceived Stress"

_ijerph, 2023, doi:10.3390/ijerph20021625_

Round 1
Reviewer 1 Report
The study investigated the relationship between family socioeconomic status and metal problems among adolescents. It is a very interesting topic. Although this paper is well organized and written, it would be great if authors can highlight the following concerns:
1. Authors mentioned that a total of 2144 questionnaires were distributed with a valid return rate of 97.29%. So what is the definition of valid? Could authors explain more details about consistent response?
2. How did authors exclude the confounding factor of family cluster.
3. All the participants of the study are coming from a junior high school, how did authors can expand their results and conclusions to other populations, such as in a different city or even in another high school in the same city.?
4. Did authors apply any multiple comparison corrections for lots of correlation analyses? What methods did authors use?
5. It would be great if authors can add a paragraph of limitations of the study.
Author Response
Dear Reviewer and Editor:
We feel great thanks for your professional review work and concern on our manuscript titled “Family Socioeconomic Status and Adolescents' Mental Problems during the COVID-19 pandemic: The Mediating Roles of Trait Mindfulness and Perceived Stress”. These professional comments helped us improve our manuscript and provided important guidance for future research. As you are concerned, there are several problems that need to be addressed. According to your valuable suggestions, we have made extensive corrections to our previous draft. We hope this meets your requirements for publication.
If there were other problems we should revise, please do not hesitate to contact us.
Best Regards,
Corresponding Author

Reviewer 2 Report
Overall, this is interesting research exploring relationships among variables that provide greater understanding about adolescent behavior and emotional wellbeing.
Here are a few suggestions for the authors to consider to provide greater clarity to the research presented.
The study uses the term "mental problems" throughout. From many perspectives, this is a pejorative term. Use of the term mental health problem would perhaps be more acceptable.
Line 36- it is stated that little is known about family SES and mental problems among adolescents. This statement should be clarified-Are the authors referring to all adolescents or Chinese adolscents? A citation should be provided to support this statement.
Line 45- it may be more accurate to state that adolescents frequently engage in... as opposed to stating experience
Line 102- would suggest changing "mental problems" to mental health problems
Line 127-Description of Participants- Please clarify how the questionnaires from Sept 2021 and Novemeber 2022 are linked, if they are? Were the variables of interest collected from the same participants at two different times?
Line 177- change the heading to Mental Health Problems: Anxiety or Just state Anxiety. The same is recommended for heading for Depression (Line 183) and Externalizing Problems (line 188).
Line 211-228 essentially repeat what is in Table 2. It would be more meaningful if lines 211 to 228 provided an overall summary of the trends seen in the data. For instance, SES is positively correlated with MAAS but negatively correlated with PSS, Anxiety, Depression, and Externalizing Problems (EP). The correlations are rather small. In contrast, the correlations between MAAS and PSS, Anxiety, Depression ,and EP are negative and much larger, suggesting MAAS may have mediating effect. As expected, the correlations between PSS, Anxiety, Depression, and EP are modest and positively correlated.
It would be helpful for the authors to provide a statement about rationale for selecting Path Analysis and tying that to the test of the hypotheses.
Line 229-246- are there any overall summary of the results -trends in the results that could be stated?
Line 248- suggest "examine" instead of exam
Line 303- mental health problems instead of mental problem
Discussion section: Are the results just generalizable to just Chinese families and adolescents or do they apply beyond the study sample and context? Are the results unique to the study sample and context of COVID in China? Is there anything to be stated to the cultural aspects of Chinese family life and development in understanding the results?
line 354- suggest using " to engage in" rather than "to take"
Conclusion section- What suggestions and recommendations should be in place for mental health professionals? What suggestions can the authors provide for training mental health professionals for screening and assessment of mental health problems of adolescents around SES?
Lastly, in addition to addressing bias (common method biases test) in data collection, are there any other limitations of the study design that should be mentioned which may be a factor in the explaining the results or how the results should be interpreted and used.
Author Response

(The authors gave the same response as above.)
